# Mutations in the *Staphylococcus aureus* Global Regulator CodY confer tolerance to an interspecies redox-active antimicrobial

**Anthony M. Martini**, **Sara A. Alexander, Anupama Khare** *

Laboratory of Molecular Biology, Center for Cancer Research, National Cancer Institute, National Institutes of Health, Bethesda, Maryland, USA

* anupama.khare@nih.gov

**Data availability statement:** The whole genome and RNA sequencing data have both been deposited at NCBI Short Read Archive (SRA) associated with BioProject PRJNA1122578.

**Funding:** This work was supported by funding from the Intramural Research Program of the NIH, National Cancer Institute, Center for

## Abstract

Bacteria often exist in multispecies communities where interactions among different species can modify individual fitness and behavior. Although many competitive interactions have been described, molecular adaptations that can counter this antagonism and preserve or increase fitness remain underexplored. Here, we characterize the adaptation of *Staphylococcus aureus* to pyocyanin, a redox-active interspecies antimicrobial produced by *Pseudomonas aeruginosa*, a co-infecting pathogen frequently isolated from wound and chronic lung infections with *S. aureus*. Using experimental evolution, we identified mutations in a conserved global transcriptional regulator, CodY, that confer tolerance to pyocyanin and thereby enhance survival of *S. aureus*. A pyocyanin tolerant CodY mutant also had a survival advantage in co-culture with *P. aeruginosa*, likely through tolerance specifically to pyocyanin. The transcriptional response of the CodY mutant to pyocyanin indicated a two-pronged defensive response compared to the wild type. First, the CodY mutant strongly suppressed metabolism by downregulating core metabolic pathways , especially translation-associated genes, upon exposure to pyocyanin. Metabolic suppression via ATP depletion was sufficient to provide comparable protection against pyocyanin to the wild-type strain. Second, while both the wild-type and CodY mutant strains upregulated oxidative stress response pathways upon pyocyanin exposure, the CodY mutant overexpressed multiple stress response genes compared to the wild type. We determined that catalase overexpression was critical to pyocyanin tolerance as its absence eliminated tolerance in the CodY mutant and overexpression of catalase was sufficient to impart tolerance to the wild-type strain against purified pyocyanin and in co-culture with WT *P. aeruginosa*. Together, these results suggest that both transcriptional responses of reduced metabolism and an increased oxidative stress response likely contribute to pyocyanin tolerance in the CodY mutant. Our data thus provide new mechanistic insight into adaptation toward interbacterial antagonism via altered regulation that facilitates multifaceted protective cellular responses.

Cancer Research to AK, AMM, and SAA. The funders had no role in study design, data collection and analysis, decision to publish, or preparation of the manuscript.

**Competing interests:** The authors have declared that no competing interests exist.

## Author summary

The pathogenic bacterium *Staphylococcus aureus* frequently co-infects patients with another pathogen, *Pseudomonas aeruginosa*, and the simultaneous presence of both is associated with worse disease progression. *P. aeruginosa* produces the antimicrobial pyocyanin (PYO) during infection that is toxic to both human cells and other microbes. Since these pathogens can coexist for years within patients, we hypothesized that *S. aureus* could develop tolerance to PYO. Using experimental evolution, we selected for PYO tolerant isolates and found that they had common mutations in a gene encoding a global regulator, *codY*. We determined that a *codY* mutation enhanced survival both upon PYO exposure, and in co-culture with *P. aeruginosa*. Further, the *codY* mutation provided protection against hydrogen peroxide, which was likely the main stress induced by PYO. Transcriptional analysis indicated that the *codY* mutant decreased expression of metabolic genes and increased expression of stress-responsive genes compared to the wild-type (WT) in the presence of PYO. Artificially depleting ATP to reduce metabolism or increasing protection against peroxides were both able to protect WT from PYO-mediated killing. Thus, our study sheds light into how pathogens can increase survival in the presence of interspecies stresses via mutations in a major regulator leading to multiple protective changes in cellular state.

## Introduction

Microorganisms commonly live in the presence of other microbial species, whether in diverse environmental niches or in association with a host [1–3]. These polymicrobial communities can be structurally and functionally dynamic in part through the balance of cooperative and competitive interactions among members [4,5]. Through these interactions, microbial species can impact the fitness, behaviors, and adaptation of other constituent members of the community [6–8]. Notably, antimicrobial effects of several secreted compounds have been shown to mediate interbacterial antagonism *in vitro*, enhancing the relative fitness of the producing species [9,10]. How community members may adapt to these antagonistic interactions is, however, less well-characterized.

The potential role of microbial interactions in human disease is being increasingly appreciated [11]. In particular, the prevalence of *Staphylococcus aureus* and *Pseudomonas aeruginosa* co-infection in wounds [12] and in the airways of people with cystic fibrosis (CF) [13] has prompted extensive work characterizing the molecular interactions between these two pathogens [14–18]. In CF, simultaneous culture of *S. aureus* and *P. aeruginosa* from sputum is associated with more deleterious clinical characteristics compared to mono-infection with either pathogen in some cohorts [19–21] but not others [22–24]. Interestingly, although *P. aeruginosa* rapidly eradicates *S. aureus* under typical *in vitro* conditions [15,25], co-colonization with both pathogens *in vivo* can persist for years [26]. This suggests that one or both species likely exhibit altered physiology and/or spatial partitioning *in vivo*, and adaptations may further contribute to their co-existence.

*P. aeruginosa* virulence is largely attributable to the many toxins it produces [27]. Among these toxins is the redox-active secondary metabolite, pyocyanin (PYO) [28,29]. PYO has been detected in secretions produced during ear infection [30] and the sputum and large airways of people with CF [31] where it can contribute to cellular toxicity [28,32]. In addition to virulence, PYO is known to have antimicrobial properties against several other microbial species via the production of reactive oxygen species or inhibition of the electron transport chain (ETC) [33–35].

Bacteria can adapt to the presence of antimicrobials by evolving resistance, where they grow in higher concentrations of the antimicrobial, or tolerance, where they survive in higher concentrations of the antimicrobial [36,37]. Previous studies have identified adaptations leading to PYO resistance in *Escherichia coli*, likely by reducing intracellular PYO concentrations and altering metabolism [38], and in *Agrobacterium tumefaciens* by altering ETC function and increasing the oxidative stress response, although other mechanisms likely also play a role [39]. In *S. aureus*, it has been shown that PYO resistance can be conferred by mutations in respiratory chain components and via putative quinone resistance responses [40–42], but additional mechanisms, especially of PYO tolerance, remain unknown.

In this study we investigate the ability of *S. aureus* to adapt to the bactericidal effects of PYO using experimental evolution, identifying novel mechanisms of PYO tolerance. In our evolved populations and isolates, we observe ubiquitous mutations in CodY, a conserved transcriptional regulator of virulence and metabolism in gram-positive bacteria [43]. The breadth of mutations observed in *codY* are likely to reduce CodY activity and we show that CodY loss-of-function confers enhanced survival during treatment with PYO, and co-culture with *P. aeruginosa*. Transcriptional analysis indicates a strong response to reactive oxygen stress during PYO treatment in both the wild-type and a CodY mutant. However, we observe that in the presence of PYO, loss of CodY activity both suppresses translation-associated gene expression and produces a stronger oxidative stress response compared to the WT. Finally, we demonstrate that recapitulating these phenotypes individually via metabolic suppression through ATP depletion or the overexpression of hydrogen peroxide-detoxifying catalase, protects WT cells from PYO-mediated cell death, indicating that both these mechanisms likely underlie the enhanced PYO survival of a CodY mutant. Thus, mutations in a global regulator can fine-tune the transcriptional landscape to enable a multidimensional adaptive response to interspecies toxins.

## Results

### Experimental evolution selects for PYO tolerance in *S. aureus*

We first determined the bactericidal effect of PYO on the *S. aureus* strain JE2 by quantifying survival of exponential phase cells upon treatment with a range of PYO concentrations. PYO production by *P. aeruginosa* can range from 0 to ~30 μg/mL (~150 μM) depending on medium composition and infection site [31,44] although higher concentrations have been observed [30,45] and are likely to exist in local microenvironments. We tested PYO concentrations that covered this range and observed a concentration-dependent effect of PYO on *S. aureus* cell density, including moderate growth reduction at 12.5 and 25 μM, growth inhibition at 50 and 100 μM, and killing at 200 and 400 μM (**Fig 1A**).

To identify potential adaptations that increase survival of *S. aureus* upon PYO exposure, we decided to use experimental evolution of cells via repeated exposures to 200 μM PYO – the lowest bactericidal concentration identified. We evolved *S. aureus* by treating early exponential phase cells with PYO for 20 hours, recovering the surviving cells overnight in medium without PYO, and repeating this process over several iterations (**Fig 1B**). During the evolution, the populations did not grow in PYO, and we estimate that each recovery phase consisted of 6-13 generations. Because we observed loss of cell viability with this concentration of PYO and recovered surviving cells, our expectation was that we would identify mutants exhibiting increased survival during treatment with PYO. We evolved two independent populations ('A' and 'B') to identify mutations specific to PYO tolerance, as these mutations should be common to both evolved populations. As the number of treatments increased, we observed enhanced survival of both independent populations when treated with PYO (**Fig 1C**).

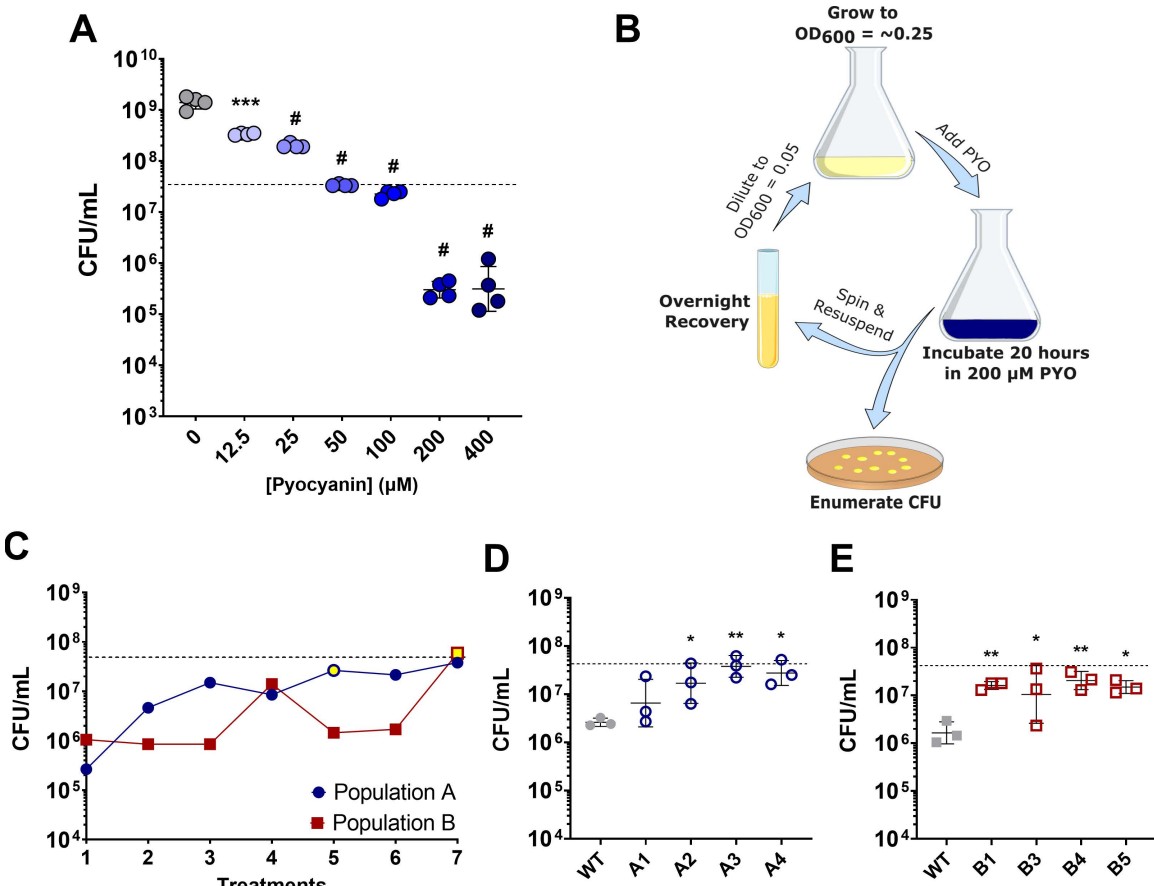

**Fig 1. Experimental evolution of *S. aureus* selects for enhanced survival in PYO.** (**A**) Early exponential phase *S. aureus* cells were treated with the indicated concentration of PYO. Values indicate *S. aureus* viable cell counts after 20 hours of PYO exposure. Data shown are the geometric mean ± geometric standard deviation of four biological replicates. (**B**) Schematic of experimental evolution to select for PYO-tolerant *S. aureus*. Overnight cultures were grown to early exponential phase prior to addition of 200 μM PYO. After incubation for 20 hours, viable cell counts were enumerated, and the remaining culture was grown overnight in the absence of PYO. This process was repeated six times for two independent populations. (**C**) Viable cell counts of two independently evolved populations, after each treatment with 200 μM PYO. Yellow fill indicates the passage at which isolates were selected. (**D, E**) Selected isolates (indicated on the x-axis) from populations (**D**) 'A' or (**E**) 'B' were assayed for tolerance to 200 μM PYO. Data shown are the geometric mean ± geometric standard deviation of three biological replicates. (**A, C, D, E**) The dashed line indicates the mean initial cell density (CFU/mL) for all strains at the time of addition of PYO or DMSO as a control (0 μM PYO). Significance is indicated for comparison to the DMSO control (0 μM PYO) (**A**), or WT (**D, E**) as determined by a one-way ANOVA using Dunnett's correction for multiple comparisons (* $P < 0.05$, ** $P < 0.01$, *** $P < 0.001$, # $P < 0.0001$).

Individual isolates from the evolved populations (designated A1, A2, B1, etc.) also exhibited higher survival following PYO exposure, indicating that these evolved strains had acquired increased tolerance to *P. aeruginosa*-derived PYO (**Figs 1D**, **1E** and **S1 Fig**). The selection of strains exhibiting reduced cell death, rather than a continuation of growth, during treatment with PYO suggested that we were selecting for PYO tolerance, instead of resistance.

## Loss-of-function mutations in the CodY global regulator confer tolerance to PYO

Next, we sought to identify common mutations that characterize PYO-tolerant isolates from each evolved population, and thus sequenced and analyzed genomes from 14 terminal isolates (7 isolates each from populations A and B). Due to the diversity of mutations observed in

the first 7 isolates of population A, we selected an additional 4 for sequencing, bringing the total to 18 sequenced isolates. While we observed diverse mutations among different isolates (S1 Data), each of the 18 isolates had at least one mutation associated with the *codY* gene (S1 Table), which encodes a well-characterized pleiotropic transcriptional regulator conserved across gram-positive bacteria [43]. In *S. aureus*, CodY regulates the expression of virulence and metabolic genes in response to nutritional cues [46,47]; however, a role in modulating tolerance to interspecies antimicrobials has not, to our knowledge, been described. Coding sequence mutations observed in our evolved isolates were present in both the substrate-sensing and DNA-binding domains, and we also observed mutations in the promoter region and the start codon (**Fig 2A** and S1 Table). Considering the diversity of mutations observed in our evolved isolates, we expect that these mutations arose during the recovery phase of the evolution and were selected for during PYO treatment. Isolation of the R61K mutation, which has previously been described to substantially reduce CodY activity [46], and an ablated start-codon, as well as the diversity of mutations across both functional domains suggested that the mutations we observed in our evolved isolates likely resulted in loss of CodY function.

To determine if a CodY mutation is sufficient to recapitulate the PYO tolerance phenotype of our evolved isolates, we reconstructed one of our evolved alleles CodY$^{R222C}$ (hereafter referred to as *codY**), in the parental strain. When the *codY** mutant was treated with 200 μM PYO, we observed significantly greater survival (~100-fold) compared to the WT (**Fig 2B**), while no difference was seen upon exposure to the DMSO control, suggesting that the *codY*

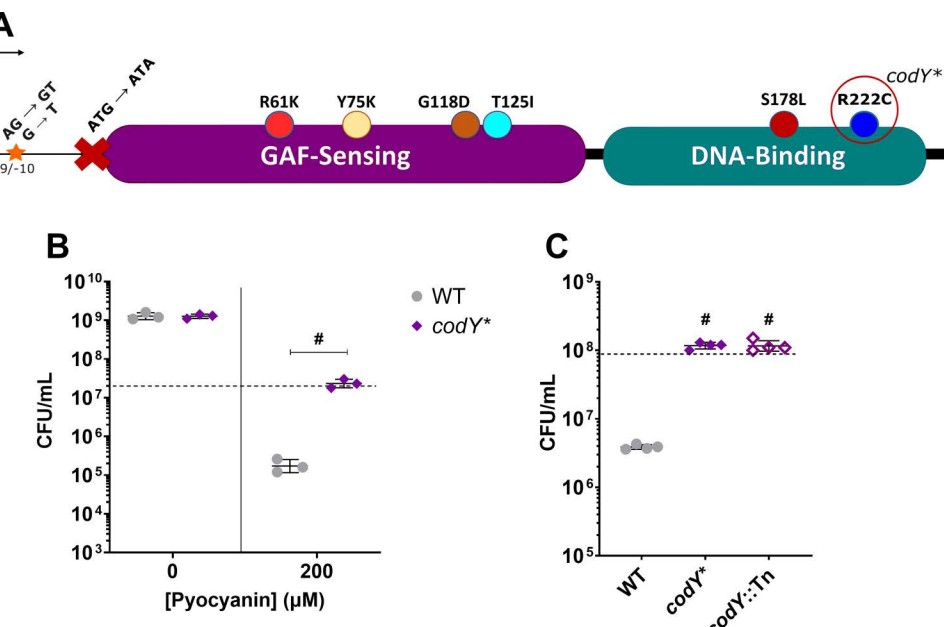

**Fig 2. Loss of CodY function confers PYO tolerance.** (**A**) Schematic of CodY-associated mutations observed in the evolved isolates and where they reside within the protein and upstream sequence. The CodY$^{R222C}$ allele is recapitulated in the *codY** mutant. (**B**) Viable cell counts of the WT and *codY** mutant after 20 hours of treatment with either DMSO as a control or 200 μM PYO. Data shown are the geometric mean ± geometric standard deviation of three biological replicates. (**C**) Viable cell counts of the WT, *codY** mutant, and a transposon mutant of *codY* (*codY*::Tn) after 20 hours of treatment with 200 μM PYO. Data shown are the geometric mean ± geometric standard deviation of four biological replicates. (**B, C**) Dashed lines indicate the mean initial cell density (CFU/mL) for all strains at the time of PYO or DMSO addition. Significance is shown for comparison to the respective WT condition as tested by a (**B**) two-way ANOVA using Tukey's correction or (**C**) one-way ANOVA using Dunnett's correction for multiple comparisons (# $P < 0.0001$).

mutations were selected for due to the PYO exposure during the evolution, and likely did not cause adaptation to just the media conditions. In addition, a mutant with a transposon insertion in *codY* knocking out CodY activity (from the Nebraska Transposon Mutant Library [NTML] [48]) phenocopied the *codY\** mutant when treated with PYO (**Fig 2C**), providing further evidence that loss of CodY function mediates PYO tolerance. The survival of the *codY\** mutant was similar to that of the evolved isolates (**Figs. 1D** and **1E**), indicating that *codY* mutations in these isolates likely cause the PYO tolerance phenotype.

We also tested whether loss of CodY function confers resistance to PYO, allowing for growth at higher PYO concentrations. We found that the *codY\** mutant exhibited greater growth than the WT at relatively low concentrations of PYO (12.5 and 25 μM), while a PYO concentration of 50 μM almost completely inhibited growth for both the WT and the *codY\** mutant (S2 Fig). These data suggest that while a *codY\** mutation confers some resistance to low concentrations of PYO, selection of *codY* mutations in the experimental evolution was largely due to increased survival and tolerance to 200 μM PYO. In addition, because we observed this moderate enhancement of PYO resistance in the *codY\** mutant (S2B and S2C Figs, **Fig 2C**), we tested whether a previously identified PYO resistance mutation would also engender increased PYO tolerance. It has been shown that loss of QsrR, a quinone-sensing repressor of quinone detoxification genes [49], results in PYO resistance in *S. aureus* [40], allowing for enhanced growth in up to 32 μM PYO compared to the parental strain. While our experimental conditions and growth medium are different, we observed greater sensitivity of the Δ*qsrR* mutant to 50 – 200 μM PYO compared to the WT (S3 Fig), indicating that although loss of QsrR allows for increased growth in the presence of lower concentrations of PYO, it does not confer increased survival or tolerance to higher concentrations of PYO in our assay conditions. Together, these data indicate that CodY loss-of-function mutations enhance *S. aureus* survival in the presence of high concentrations of PYO via a mechanism distinct from previously identified adaptive mutations.

### The *codY\** mutation confers protection against PYO-dependent killing in co-culture with *P. aeruginosa*

*P. aeruginosa* is known to produce a variety of additional molecules that may have antimicrobial effects, including against *S. aureus* [15]. We tested whether the *codY\** mutation conferred increased tolerance to several such *P. aeruginosa* antimicrobials. We saw either no difference or a minor increase (1.5- to 2-fold) in survival or growth between WT and the *codY\** mutant in the presence of the phenazines 1-hydroxyphenazine, or phenazine-1-carboxylic acid, the alkyl quinolones PQS, or HHQ, the biosurfactant rhamnolipids, or the siderophores pyoverdine, or pyochelin (S4A-S4C, S4E Fig). However, we did observe an approximately 10-fold increase in survival of the *codY\** mutant when treated with HQNO (S4D Fig), an alkylquinolone that also inhibits respiration in *S. aureus* [50,51], indicating some overlap in tolerance to another *P. aeruginosa* toxin.

Further, to determine the effect of the *codY\** mutation in a co-culture of *S. aureus-P. aeruginosa*, we first tested whether PYO and other phenazines can contribute to competitive killing of *S. aureus*. We performed co-culture experiments of WT *S. aureus* with the *P. aeruginosa* strain PA14 and isogenic mutants deficient for either both phenazine biosynthesis operons *phzA1-G1* and *phzA2-G2* (Δ*phz1/2*) or just PYO (Δ*phzM*) in media modified to enhance PYO production [44]. When co-cultured with the parental *P. aeruginosa* PA14 strain, we observed an approximately 10-fold decrease in *S. aureus* survival (**Fig 3A**), that was not seen in co-culture with either the Δ*phz1/2* or Δ*phzM* mutants (**Fig 3A**), suggesting that PYO may be a major contributor to *S. aureus* killing in these conditions. Finally, we observed that the *codY\** mutation was protective in co-culture with the parental *P. aeruginosa* strain compared to WT

*S. aureus* (**Fig 3B**). Together, these data suggest that *codY** contributes to *S. aureus* survival against PYO-dependent killing in co-culture with *P. aeruginosa*.

## A CodY[R222C] mutant exhibits expected transcriptional changes in metabolism and virulence gene expression

CodY regulates a substantial proportion of the *S. aureus* genome based on branched-chain amino acid (isoleucine, leucine, and valine) and nucleotide (GTP) availability [47,52]. In the presence of sufficient intracellular concentrations of these nutrients CodY is active and functions primarily to repress its target genes [46]. As nutrients become scarce, CodY activity decreases, thereby facilitating the expression of amino acid transport and biosynthesis genes and host-targeting virulence factors.

To identify differentially expressed genes between the two strains that may explain the increased PYO tolerance of the *codY** mutant, we determined the transcriptional response of the WT and *codY** mutant to PYO at early (30 minutes) and late (120 minutes) time points (S2 Data). The gene expression changes in the *codY** mutant compared to WT under control conditions at 30 minutes were consistent with previous reports of the CodY regulon [46,47] (S5 Fig and S2 Data) indicating that CodY[R222C] disrupts CodY-dependent regulatory activity, similar to a *codY* null mutant. We observed no difference in gene expression at 120 minutes, likely due to the natural de-repression of CodY at this time point in the WT strain.

In addition, in the *codY** mutant we observed overexpression of the *agr* quorum sensing system (S5B Fig) that regulates virulence gene expression [53,54], and can impart long-lived protection from oxidative stress [55]. However, we did not observe enhanced PYO sensitivity of an *agrA*::Tn mutant and a *codY* agrA*::Tn double mutant exhibited protection against PYO comparable to the *agr* competent strains (S6 Fig), suggesting that overexpression of *agr* in the *codY** mutant is not responsible for increased PYO tolerance.

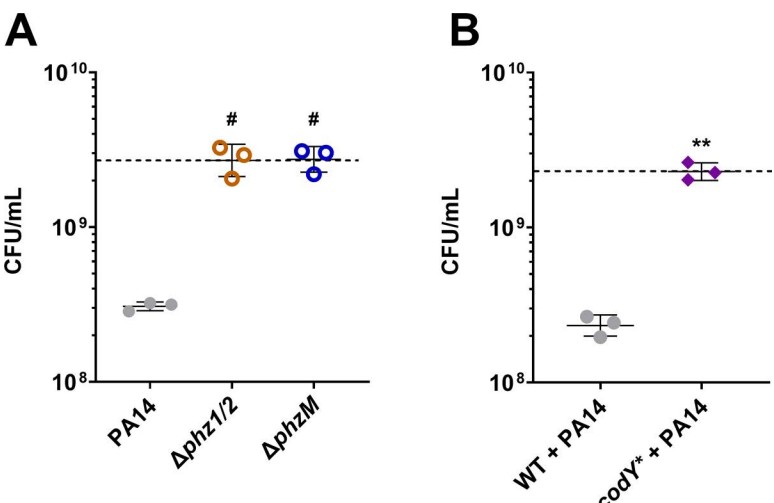

**Fig 3. The *codY** mutant is tolerant to PYO-dependent killing by *P. aeruginosa* in co-culture.** (**A**) Viable cell counts of *S. aureus* JE2 after 18 hours of co-culture with *P. aeruginosa* WT, a phenazine-null mutant (Δ*phz1/2*), or a PYO-null mutant (Δ*phzM*). (**B**) Viable cell counts of *S. aureus* WT or the *codY** mutant after 18 hours of co-culture with *P. aeruginosa* PA14. (**A, B**) Data shown are the geometric mean ± geometric standard deviation of three biological replicates and the dashed lines indicate the mean initial cell density of all *S. aureus* strains. Significance is shown for comparison to the (**A**) PA14 condition or (**B**) WT condition as tested by (**A**) one-way ANOVA using Dunnett's correction for multiple comparisons or (**B**) unpaired, two-tailed t-test (# $P < 0.0001$).

### *S. aureus* responds to PYO by inducing stress response and iron acquisition genes and suppressing metabolism

Considering that *S. aureus* and *P. aeruginosa* are frequently found to co-infect patients, and *S. aureus* may thus be exposed to PYO, we determined the response of the WT *S. aureus* JE2 strain to PYO. Enriched pathways among the PYO-induced genes included iron acquisition (siderophore and heme metabolism), energy generation (menaquinone biosynthesis and the TCA cycle), and stress response to oxidative stress, metals, and DNA damage (**Fig 4**), indicating that PYO likely generates reactive oxygen species that are known to affect metal homeostasis [56–58], causes DNA damage, and leads to metabolic alterations in *S. aureus*. While the *codY\** mutant overall showed a similar response to PYO (S7 and S8 Figs), at the early time

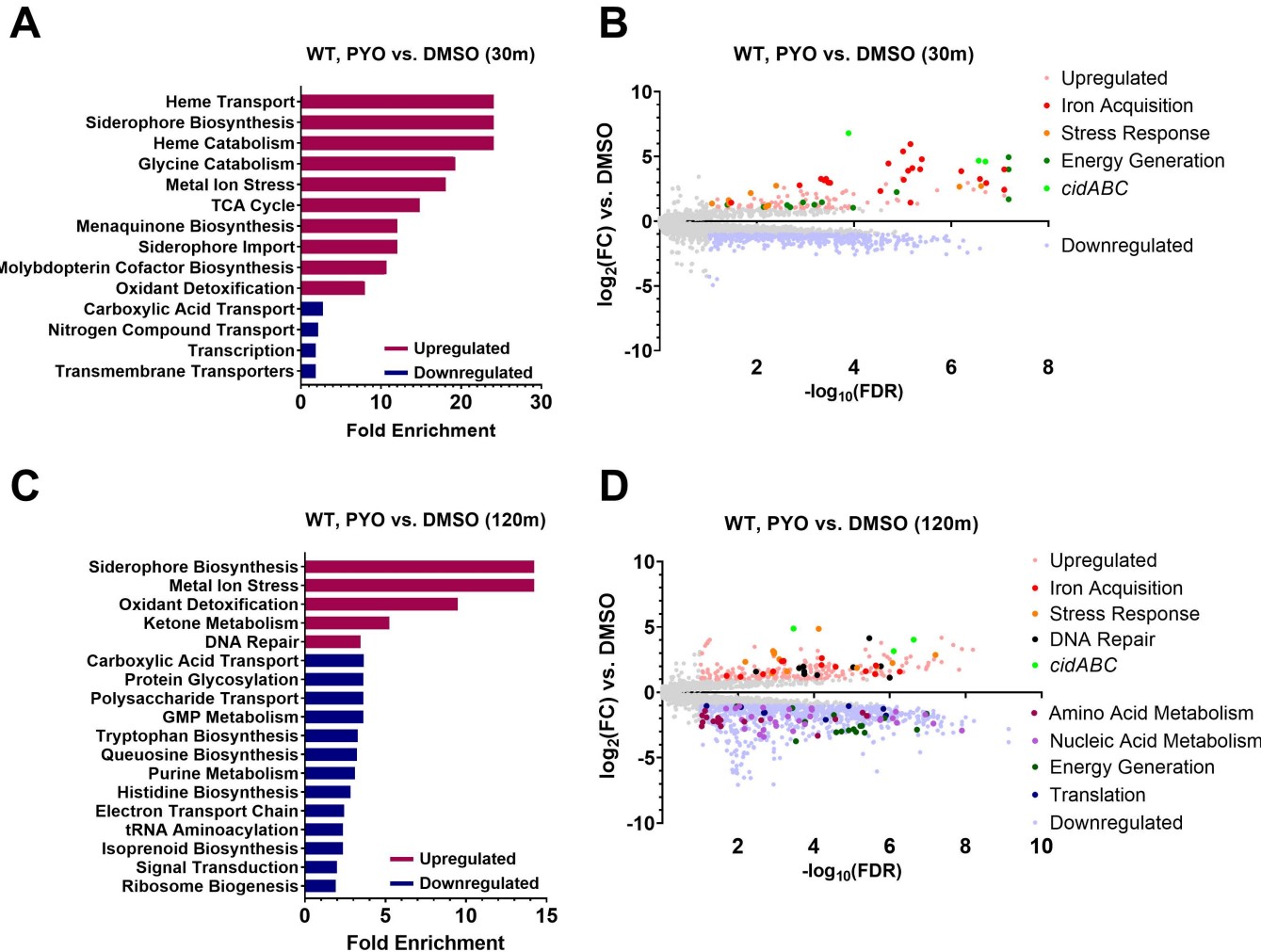

**Fig 4. S. aureus increases expression of genes associated with iron acquisition and stress responses while suppressing metabolism- and translation-related genes in response to PYO.** Differential gene expression of WT *S. aureus* in response to 200 μM PYO after 30 (**A, B**) and 120 (**C, D**) minutes of incubation in PYO. (A, C) Enriched GO pathways at 30 and 120 minutes, respectively, among upregulated and downregulated genes and their fold enrichment relative to the expected number of observed genes. (**B, D**) Volcano plots of log$_2$(fold change gene expression) and -log$_{10}$(false discovery rate) at 30 and 120 minutes, respectively. Upregulated genes are shown in light red and downregulated genes are shown in light blue. Further highlighted genes indicate the over- and under-expressed genes comprising the associated functional pathways in the legend (see S3 Data for a list of the included genes). Enriched GO pathways (**C**) and differentially expressed genes (**D**) at 120 minutes include only those not also observed in the *codY\** mutant compared to WT from the 30-minute DMSO comparison (S5 Fig), but a full list can be found in S8 Fig and as part of S2 Data.

point we observed broad repression of amino acid and nucleic acid metabolism, energy generation, and translation-associated pathways (S7A and S7B Figs), suggesting that PYO rapidly elicits a more metabolically dormant state in the *codY** mutant, a phenomenon that has been associated with antibiotic tolerance and persistence [36,59].

We also noted that *cidA*, part of the *cidABCR* operon that plays a central role in controlling carbon metabolism and programmed cell death [60,61], was the most overexpressed gene in the WT JE2 strain at both early (~111-fold) and late (~29-fold) time points, while *cidB* and *cidC* were also highly upregulated (**Fig 4B and 4D**). We therefore tested whether any of the *cid* genes were involved in PYO-dependent cell death or tolerance. We observed no effect for mutations in any of the *cidABCR* genes in either the WT or *codY** backgrounds (S9 Fig), suggesting that, despite its strong overexpression, the *cid* operon does not contribute to cell death or survival in these conditions.

Since we also observed enrichment of DNA repair pathways among upregulated genes, we determined if PYO treatment led to DNA damage using a TUNEL assay. We observed significant DNA damage after 2-, 4-, and 20-hour treatment with PYO in the WT strain (S10A Fig). Further, while WT exhibited a moderate, but not statistically significant, increase in TUNEL staining compared to the *codY** mutant in PYO at 2 hours, there was no significant difference at other time points (S10B **and** S10D-F Figs). Since we do not observe killing following 2- and 4-hour treatment with PYO (S10C Fig), these data suggest that while PYO does induce DNA breaks, these likely do not explain the differential PYO tolerance between WT and the *codY** mutant.

## The *codY** mutant exhibits greater suppression of translation and expression of stress response genes than the WT

We next directly compared gene expression between the *codY** mutant and WT in the PYO condition to identify transcriptional differences that could potentially explain the PYO tolerance of the *codY** mutant. Apart from CodY-dependent amino acid metabolism pathways, at the early time-point, translation was enriched among the downregulated genes (Fig 5A and S11A Fig), consistent with the greater metabolic suppression seen in the *codY** mutant at this time-point. Further, several genes and pathways potentially involved in stress responses were upregulated in the *codY** mutant. These included *pxpBCA* encoding 5-oxoprolinase that converts 5-oxoproline to glutamate [62], and *gltBD* encoding glutamate synthase, both of which generate glutamate that has been implicated in the response to oxidative and other stresses [63–65], and *adhC* encoding alcohol dehydrogenase, that has been implicated in resistance to oxidative and nitrosative stress [66,67], which were consistently upregulated at both time points (**Fig 5A and 5B**). Additionally, the carotenoid biosynthesis pathway, and catalase-encoding *katA*, which are both important for resistance to oxidative stress [56,68] were upregulated at the later time-point (**Figs 5B** and S11 Fig) suggesting that the *codY** mutant has a more robust stress response to PYO which may underlie its increased survival.

## ATP depletion protects *S. aureus* from the bactericidal effects of PYO

Given the increased metabolic suppression in the *codY** mutant upon PYO exposure, and that ATP depletion in *S. aureus* and *E. coli* is associated with the formation of persister cells and reduced susceptibility to antibiotics [69,70] we hypothesized that metabolic quiescence may promote protection from PYO. We therefore tested the effect of proton motive force decoupling agents carbonyl cyanide m-chlorophenyl hydrazone (CCCP), a protonophore that reduces ATP synthase activity [71,72], or sodium arsenate, which reduces ATP by forming unproductive ADP-arsenate [69,73], on PYO tolerance. Treatment with either 10 µM CCCP

or 30 mM arsenate resulted in WT PYO tolerance comparable to that of the *codY\** mutant (**Fig 6A**). We observed no additional protective effect of ATP depletion via addition of 10 μM CCCP on PYO tolerance of the *codY\** mutant (S12 Fig). While CCCP did not interfere with growth in the absence of PYO, arsenate arrested growth at the point of addition.

Despite their metabolic dormancy, persister cells maintain some active metabolic processes [74,75] which can be necessary for the protective effects of persistence [76–78]. To determine whether CCCP-mediated protection from PYO still required active metabolism, we tested the effect of higher CCCP concentrations on both the WT and *codY\** mutant strains. While 40 μM CCCP did not adversely affect growth in the DMSO control, this concentration resulted in an increased susceptibility to PYO in both the WT and *codY\** mutant strains (S12 Fig), suggesting that active metabolism is likely required for basal PYO tolerance in the WT and increased PYO tolerance in the *codY\** mutant. Together, these data indicate that partial metabolic suppression in the *codY\** mutant may contribute to PYO tolerance.

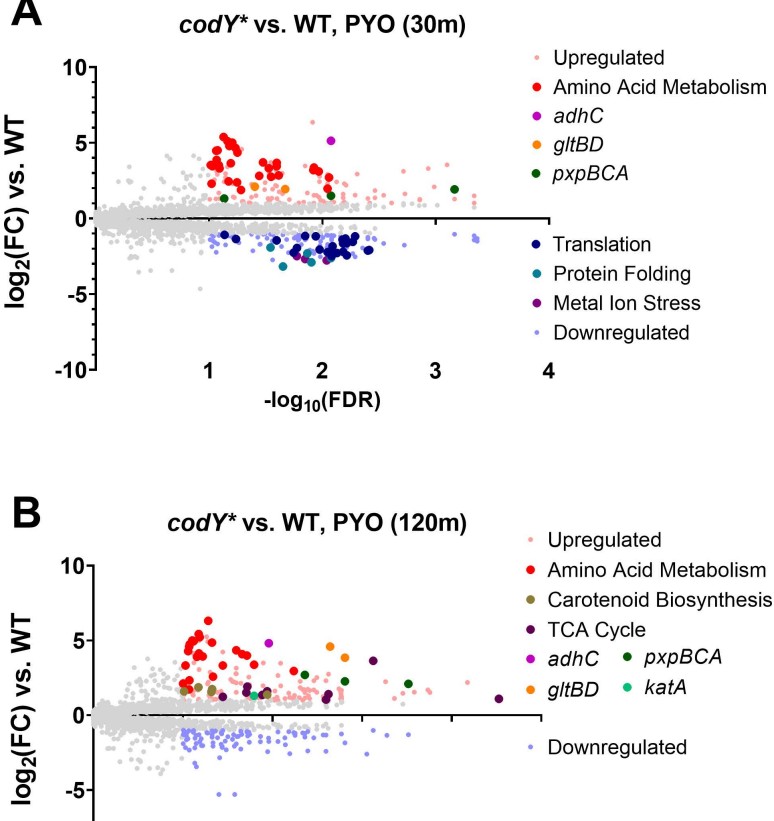

**Fig 5. The *codY\** mutant exhibits enhanced expression of amino acid metabolism and stress response genes and downregulates translation compared to the WT in response to PYO.** Volcano plots showing $\log_2$(fold change gene expression) and -$\log_{10}$(false discovery rate) in response to 200 μM PYO in the *codY\** mutant compared to WT after 30 (**A**) and 120 (**B**) minutes. Highlighted genes indicate the over- and under-expressed genes comprising the associated functional pathways (see S3 Data for a list of the included genes) or selected genes associated with stress responses. Enriched GO pathways for the respective volcano plots can be found in S11 Fig.

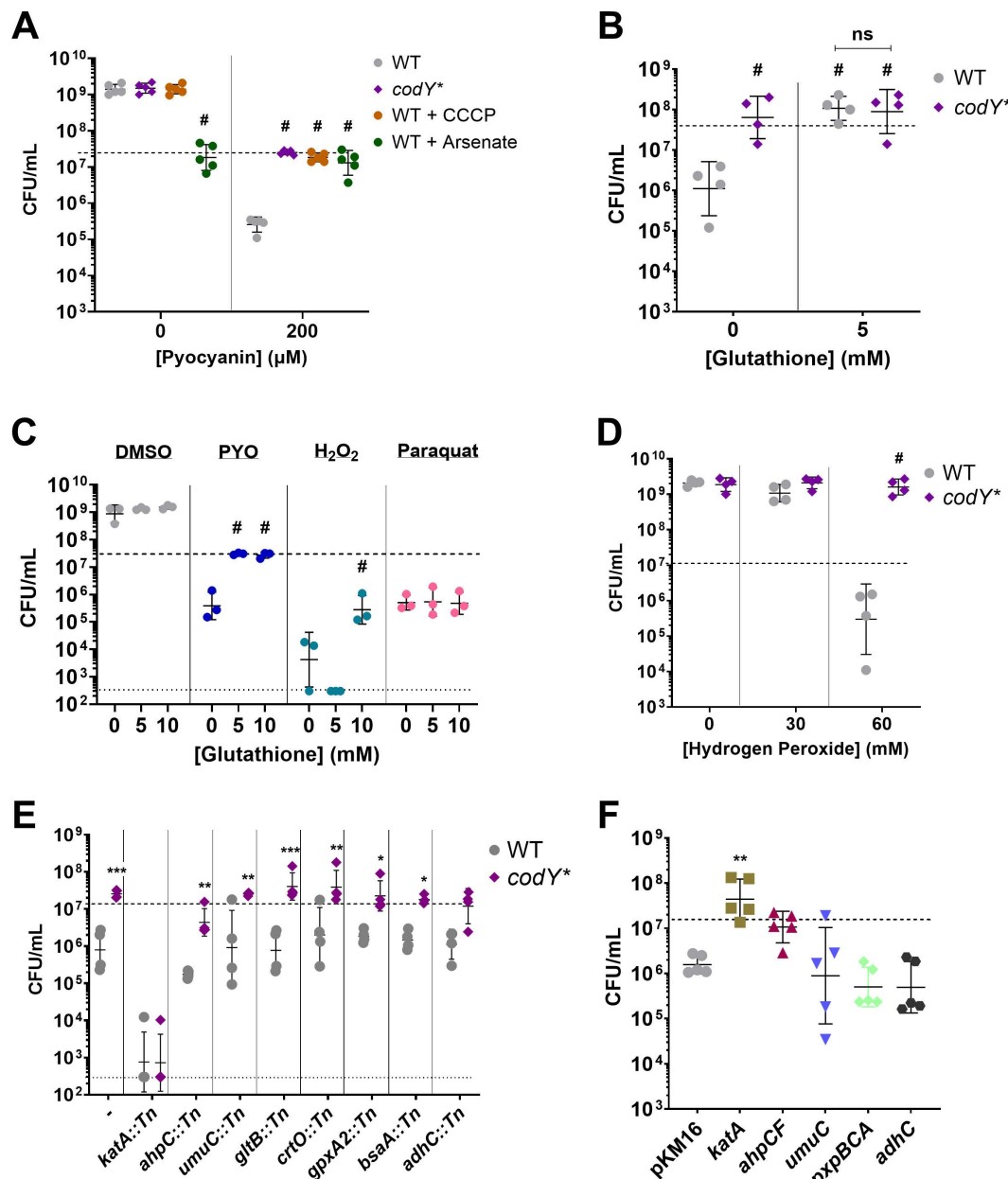

**Fig 6. Metabolic suppression and hydrogen peroxide stress response mediate PYO tolerance.** Viable cell counts of the indicated *S. aureus* strains after 20-hour treatment with the stated reagents. (**A**) Tolerance to 200 µM PYO of the *S. aureus* *codY\** mutant, and WT in the presence of the ATP depleting agents CCCP (10 µM) and sodium arsenate (30 mM). (**B**) Tolerance to 200 µM PYO of the WT and the *codY\** mutant in the presence of glutathione or sterile water as a control. (**C**) Effect of glutathione on WT survival during treatment with 200 µM PYO, 60 mM hydrogen peroxide, or 0.1 mM paraquat. (**D**) Susceptibility of WT and the *codY\** mutant to hydrogen peroxide. (**E**) PYO sensitivity of transposon mutants disrupting overexpressed genes from RNA-seq in WT and the *codY\** mutant backgrounds. (**F**) Overexpression of selected genes with their native promoters from a multi-copy plasmid. pKM16 expresses the fluorescent protein dsRed3.T3 from the *sarA1* promoter of *S. aureus* and is used as a control. Supplemental reagents were added at the indicated concentrations immediately prior to addition of 200 µM PYO, DMSO, or sterile water. Dashed lines indicate the mean initial cell density (CFU/mL) for all strains prior to addition of PYO, DMSO, or other reagents, while the lower dotted lines in (**C**) and (**E**) indicate the limit of detection. Data shown are the geometric mean ± geometric standard deviation of the following numbers of biological replicates: (**A**) five, (**B**) four, (**C**) three, (**D**) four, (**E**) four, (**F**) five. Significance is shown relative to (**A**) the respective WT condition, (**B**) WT without glutathione, or the indicated strains, (**C**) the respective no glutathione condition, (**D**) the respective WT condition, (**E**) the respective WT mutant, or (**F**) pKM16, and was determined by (**A-E**) a two-way ANOVA using (**A**) Dunnett's, (**B, C, D**) Tukey's, or (**E**) Šídák's correction, or (**F**) one-way ANOVA using Dunnett's correction for multiple comparisons (\* $P < 0.05$, \*\* $P < 0.01$, \*\*\* $P < 0.001$, # $P < 0.0001$).

## Mitigation of hydrogen peroxide toxicity is protective against PYO

PYO toxicity is in part mediated by the generation of diverse ROS [33,79], and the oxidative stress response pathway is induced in both the WT and *codY\** mutant upon PYO exposure (**Figs 4** and S7 Fig). To determine whether the PYO-induced cell death observed in the WT was mediated by ROS, we screened *S. aureus* mutants deficient in different ROS responses for sensitivity to PYO. These included mutants in the catalase (*katA*) and alkyl hydroperoxide reductase (*ahpC, ahpF*) hydrogen peroxide detoxification systems [80,81], the peroxiredoxins thiol peroxidase (*tpx*) and thiol-dependent peroxidases (*bsaA, gpxA2*) predicted to detoxify peroxides [82], both superoxide dismutases (*sodA, sodM*) [83], and a biosynthesis gene for bacillithiol (*bshA*) predicted to function in redox homeostasis [82]. Mutations in *katA*, *ahpC*, and *ahpF* sensitized WT to PYO-mediated killing (S13 Fig), indicating that detoxification of hydrogen peroxide is critical for PYO tolerance. Interestingly, a mutant of *perR*, a master repressor of the oxidative stress response [56,57], phenocopied the PYO tolerance of the *codY\** mutant (S13 Fig). A *perR* mutant constitutively expresses several stress response genes, including *katA*, *ahpCF*, *dps*, *trxB*, and *ftnA* [56], corroborating that an enhanced ROS response is protective against PYO.

To further test whether ROS were responsible for the killing effect of PYO, we supplemented WT with the antioxidant glutathione [84–86], and observed substantially increased survival of the WT, comparable to the *codY\** mutant (**Fig 6B**). Similar to the CCCP treatment, we observed no additional protective effect of glutathione on the *codY\** mutant. Glutathione supplementation also provided partial protection against hydrogen peroxide, but not paraquat-induced superoxide (**Fig 6C**), indicating that hydrogen peroxide is a primary mediator of cell death under these conditions, consistent with the enhanced PYO sensitivity of the *katA* and *ahpCF* mutants (S13 Fig).

In the *codY\** mutant, *katA* was overexpressed 2.2- to 2.5-fold compared to WT in DMSO (S2 Data) and in the later response to PYO (Fig 5B), suggesting that the *codY\** mutant exhibits an elevated basal tolerance to hydrogen peroxide and a greater response to ROS stress compared to WT. We found that the *codY\** mutant exhibited increased resistance to hydrogen peroxide (**Fig 6D**) but not superoxide (S14 Fig). Together, these data indicate that PYO-mediated toxicity is primarily mediated by hydrogen peroxide and that *codY\** confers greater tolerance to this ROS.

To identify the specific genes that mediate the increased tolerance of the *codY\** mutant to ROS, and specifically PYO, we assessed several stress response related genes whose transcripts were elevated in the *codY\** mutant either upon PYO exposure or compared to the WT in response to PYO. In addition to *katA*, *ahpC*, *bsaA*, and *gpxA2*, we selected a gene in the biosynthesis pathway of carotenoids (*crtO*) which function to alleviate hydrogen peroxide stress [68,87]; alcohol dehydrogenase (*adhC*) which is part of the Rex regulon responsive to redox stress and oxygen limitation [88,89]; MucB (*umuC*), an error-prone DNA polymerase which functions in DNA repair [90]; and glutamate synthase (*gltB*) since glutamate utilization can mediate protection against oxidative stress [63]. We transduced transposon mutations in each of these genes to the *codY\** mutant background. When challenged with PYO, we found that loss of *katA* (*codY\* katA*::Tn) substantially reduced PYO tolerance and abolished the protective effects of *codY\**, suggesting that *katA* is required for PYO tolerance in the *codY\** mutant (**Fig 6E**). In contrast, while the *codY\* ahpC*::Tn strain showed lower survival compared to the *codY\** mutant, it still showed higher tolerance compared to the *ahpC*::Tn strain.

Finally, we asked whether overexpression of these or other stress-related genes would be sufficient to confer PYO tolerance. When overexpressed from a multi-copy plasmid, we observed that *katA* protected WT from PYO-mediated killing at levels similar to *codY\** and

expression of *ahpCF* exhibited a moderate, though not statistically significant, protective effect (**Fig 6F**). In contrast, overexpression of *umuC*, *pxpBCA*, or *adhC* was not sufficient to protect WT from PYO-mediated killing (**Fig 6F**).

Subsequently, we tested whether overexpression of *katA* would confer protection against *P. aeruginosa* in co-culture and similarly observed an approximately 10-fold increase in survival (S15 Fig). Taken together, these results indicate that an enhanced response to hydrogen peroxide stress is sufficient to mediate PYO tolerance and that the overexpression of *katA* in the *codY** mutant contributes to its increased survival in PYO.

## Experimentally evolved mutations in CodY are present in genomes of *S. aureus* clinical isolates

In our experimental evolution, we identified *codY* mutations in each of 18 sequenced isolates from two independently evolved populations. In total, we observed nine different mutations: seven in the coding sequence including an ablation of the start site, and two promoter mutations. Given that we did not isolate the same coding sequence mutation from both independently evolved populations, and that these mutations probably led to loss of CodY function, there is likely a large mutational space of CodY-inactivating alleles that may be selected for upon exposure to redox stress. Further, it has been shown that a *codY* deletion leads to increased *in vivo* virulence in a USA300 strain [91], suggesting that this could be an additional selective pressure for *codY* inactivating alleles.

To test whether *codY* mutations are seen in publicly available genomes of *S. aureus* strains, we queried the JE2 CodY protein sequence against a set of 63,983 *S. aureus* genomes from the NCBI Pathogen Detection Database (S4 Data). Interestingly, we identified multiple isolates which had CodY mutations that we observed in our study, including T125I, S178L, and R222C, as well as a variety of other mutations throughout the protein, suggesting that these mutations can and do arise in natural isolates (S4 Data). The mutations throughout the protein likely also reflect natural variation, and further work is required to determine the effect of such variation on CodY function, as well as adaptation to PYO and *P. aeruginosa*.

## Discussion

Constituents of polymicrobial communities can exhibit competitive behaviors that affect other community members [9]. Such selective pressures within these communities can promote adaptations that maintain or shift the balance of community form and function [4], but the breadth of these mechanisms is not well-characterized. In this study, we use experimental evolution to investigate the adaptive response of *S. aureus*, a widespread pathogen frequently identified in antibiotic-resistant and polymicrobial infections, to the redox-active antimicrobial PYO, produced by the co-infecting pathogen *P. aeruginosa*. We show that recurrent treatment with a bactericidal concentration of PYO selects for increased *S. aureus* survival mediated by loss-of-function mutations in the pleiotropic transcriptional repressor, CodY (**Fig 1C**-**1E** and S1 Table). CodY mutation alone is sufficient to confer increased survival against purified PYO (**Fig 2B** and **2C**) and *P. aeruginosa* in co-culture (**Fig 3B**), suggesting that de-repression of CodY enhances *S. aureus* fitness under these conditions. Transcriptional analysis during PYO treatment indicates that the *codY** mutant shows a stronger repression of translation-associated genes and greater expression of certain stress response genes compared to WT (**Fig 5**), suggesting that transcriptional changes in the *codY** mutant confer PYO tolerance. Consistent with this hypothesis, we observed that, individually, metabolic suppression or overexpression of catalase was sufficient to impart PYO tolerance to the WT (**Fig 6A** and **6F**). Our results suggest a multifaceted adaptive response to antimicrobial-induced reactive

oxygen stress that reduces lethal cellular damage through lowered metabolism and enhanced ROS detoxification.

PYO and other phenazines have diverse functions in *P. aeruginosa* physiology [92–96], pathogenesis [32,97], and interbacterial competition [38,39]. In our experimental system, the ability of *P. aeruginosa* to produce PYO contributed to *S. aureus* killing in co-culture (**Fig 3A**). The ability of PYO to accept and donate electrons enables it to interfere with respiratory processes of other species [38,42,98] and generate toxic ROS through the reduction of molecular oxygen [33,39]. Although PYO is frequently undetectable in CF sputum even during colonization with *P. aeruginosa* [99–101], PYO production has been observed during human disease, including in ear and CF lung infections [30,31] and several lines of evidence suggest a potential role in infection. Culture in *ex vivo* CF sputum [102], *in vitro* in CF-sputum mimicking medium [103] or in the presence of anaerobic products frequently found in CF lung environments or breath condensates [104–107] can induce expression of PYO biosynthesis genes or PYO production. Intra- and inter-species metabolic cross-feeding of citrate is also capable of stimulating PYO production [108]. In addition, overproduction of PYO [109] and regulatory rewiring that maintains PYO production [110] have been observed in CF isolates, and one study showed an association between increased PYO production and isolates from pulmonary exacerbation sputum samples [111].

Recently published studies have used experimental evolution to identify diverse adaptive mechanisms of *S. aureus* to *P. aeruginosa* antagonism. Loss-of-function mutations in the aspartate transporter, *gltT,* led to increased survival of *S. aureus* during surface-based co-culture competition with *P. aeruginosa* [112], as selective pressure under those conditions was primarily related to competition for amino acids. While we observe downregulation of *gltT* and the glutamate transporter *gltS* in response to PYO in both WT and the *codY\** mutant (S2 Data), we would not expect similar selective pressures in our assay. In a separate study, *S. aureus* evolved in the presence of *P. aeruginosa* supernatant showed strain-dependent acquisition of resistance that converged on staphyloxanthin (carotenoid) production and the formation of small colony variants (SCVs) [113]. Interestingly, in two different strains, 1 out of 5 populations each encoded a mutation in *codY*, one being intergenic and the other a non-synonymous mutation. While we also observed overexpression of staphyloxanthin biosynthesis genes (**Fig 5B**), we did not observe increased sensitivity to PYO in a *crtO* knockout mutant (**Fig 6E**). It is likely that differences in the primary phenotype (resistance versus tolerance) and the mixture of inhibitory factors present in *P. aeruginosa* supernatant can explain this difference, but also suggests that the consequences of *codY* mutation can facilitate protective phenotypes in other conditions.

Selection of *S. aureus* mutants that can grow in the presence of PYO identified SCVs and mutations in *qsrR* as PYO resistance determinants in previous studies [40,98]. The antimicrobial activity of PYO is considered to involve two functions: ETC inhibition and generation of ROS [33,34]. SCVs likely evade ETC inhibition and ROS generation due to reduced respiration. Alternatively, *qsrR*-mediated responses are reported to detoxify PYO leading to PYO resistance [40]. It seems likely that ETC inhibition was critical for the bacteriostatic effects of PYO and subsequent PYO resistance mechanisms in these studies. We did not observe SCVs in our evolution, or any SCV-associated mutations in our sequenced isolates, possibly because our evolution included a growth (recovery) phase, which would likely deplete slow-growing SCV mutants. Further, based on our results suggesting that PYO tolerance is decoupled from PYO resistance, it is possible that SCVs provide resistance to PYO but do not alter tolerance levels.

Instead, in our conditions, we observed substantial induction of stress response transcripts (**Fig 4**), suggesting that ROS generation is a major effect of PYO. In particular,

expression of *katA*, *ahpCF*, and the cytoplasmic iron-sequestering protein *dps* that protects DNA from hydrogen peroxide [114,115] were highly overexpressed in response to PYO in both the WT and *codY\** mutant (**Figs 4**, S7 and S2 Data). The *cidABCR* operon was very highly induced upon PYO exposure, possibly due to a metabolic or physiological shift following respiratory inhibition by PYO. Of note, the redox-sensing two-component regulator SrrAB is known to regulate *cidABC* expression [116,117] and could be a link between PYO exposure and *cidABC* overexpression. Additionally, genes involved in distinct iron acquisition systems were among the most upregulated genes in response to PYO (**Figs** and S7 Fig). In *S. aureus*, metal acquisition and homeostasis genes are integrated into the regulons of peroxide stress response regulators such as PerR and Fur [56,57], likely due to the iron-dependent functions of redox proteins such as catalase. Hydrogen peroxide can also induce expression of heme and iron uptake genes [58]. Together with the observation that over-expression of *katA* is sufficient to induce PYO tolerance (**Fig 6F**) and increase survival in co-culture with PYO-producing *P. aeruginosa* (**Fig 3A** and S15 Fig), these data suggest that the bactericidal effects of PYO in *S. aureus* are primarily driven by the generation of peroxides. Interestingly, this is in contrast to observations in *A. tumefaciens* where superoxide dismutase was the critical ROS stress protein mediating PYO tolerance [39]. It is possible that this distinction reflects differences between the physiology of the two organisms and the experimental conditions.

CodY regulates approximately 5-28% of the *S. aureus* genome depending on the strain and experimental conditions [47,52], predominantly by repressing genes that function in metabolism and virulence factor production [43,46]. Therefore, the presence of mutations associated with *codY* raised the immediate possibility that transcriptional changes prior to or during PYO treatment contribute to PYO tolerance. Indeed, we observed differential expression of multiple stress response genes during PYO treatment in the *codY\** mutant compared to WT (**Fig 5B**, S5B and S2 Data). Most of these genes (*katA*, *crtO*, *gpxA2*, *gltB*) are known or predicted to detoxify peroxides. However, among these overexpressed genes, only loss of *katA* fully sensitized the *codY\** mutant to PYO (**Fig 6E**). Overexpression of *katA* was also observed in the *codY\** mutant in the absence of PYO (S2 Data), likely contributing to its enhanced resistance to hydrogen peroxide (**Fig 6D**) but not superoxide (S14 Fig). Based on the protection provided by the *codY\** mutation even in an *ahpC* mutant (**Fig 6E**), it is likely that most of the toxic effects of PYO are borne from high levels of hydrogen peroxide that are more effectively detoxified by the functionally intact catalase in this mutant [81]. Based on our results, the enhanced stress response could contribute to the increased virulence of a CodY mutant [91], alongside the de-repression of virulence factors, by conferring protection from host defenses [46,47].

In addition to overexpression of stress responses, the *codY\** mutant also exhibited a strong, early reduction of translation-associated gene expression compared to the WT (**Fig 5A**), and several amino acid and nucleotide biosynthesis pathways as well as the electron transport chain were all enriched among genes downregulated early in the *codY\** mutant upon PYO exposure (S7 Fig). This indicated a rapid metabolic suppression in the *codY\** mutant by PYO and artificially depleting ATP was sufficient to protect WT from PYO-mediated killing (**Fig 6A**). Consistent with lowered metabolism leading to PYO tolerance, it has been previously shown that the development of antibiotic tolerance and persistence is mediated by metabolic restriction, where reduced translation, ATP levels, or growth rate can enhance survival against antimicrobials [69,118]. In this context, suppressing metabolism may also contribute to protection by reducing the accumulation of PYO-generated ROS. Combined with increased expression of *katA*, our results suggest that both responses contribute to PYO tolerance via overlapping mechanisms.

Notably, in our experiments, although a *perR* mutant showed high PYO tolerance (S13 Fig), and we saw induction of many genes within its regulon upon PYO exposure, *perR* mutations were not present in any of our evolved isolates. Similarly, we did not identify any promoter mutations in *katA* that would lead to overexpression. This could be due to the small number of populations we evolved, minor fitness costs associated with *katA* promoter mutations and *perR* loss of function mutations, or unique selective pressures exerted by our experimental evolution protocol. Alternatively, it is possible that the levels of *katA* induction accessible to such specific mutations are lower than what we see via plasmid overexpression, and not sufficient by itself to lead to high levels of PYO tolerance. Instead, the experimental evolution selected for *codY* mutations that led not only to *katA* overexpression, but also metabolic suppression, and the combination of both these effects likely led to the observed high PYO tolerance.

It has been suggested that mutations in regulators can facilitate adaptation by optimizing regulatory processes towards a new niche, via increasing expression of critical biochemical capabilities, or inhibiting wasteful or damaging metabolic processes [119]. Our study shows that mutations in global regulators may also allow access to unique peaks in the fitness landscape due to pleiotropic effects on cellular metabolism, thereby facilitating multiple distinct protective responses.

## Materials and methods

### Bacterial strains and growth conditions

All strains and plasmids used in this study are described in S2 Table. Bacteria were cultured at 37 °C with shaking at 300 rpm in modified M63 medium [16] (13.6 g/L $KH_2PO_4$, 2 g/L $(NH_4)_2SO_4$, 0.4 μM ferric citrate, 1 mM $MgSO_4$; pH adjusted to 7.0 with KOH) supplemented with 0.3% glucose, 1x ACGU solution (Teknova), 1x Supplement EZ (Teknova), 0.1 ng/L biotin, and 2 ng/L nicotinamide for all experiments. For the co-culture experiments (shown in **Fig 3** and S15 Fig), and experiments testing the effect of other *P. aeruginosa* antimicrobials (shown in S4 Fig), this medium was supplemented with 10 mg/mL tryptone, 2 μM ferric citrate, and 0.6% glycerol (M63T). Luria-Bertani (LB) broth (KD Medical) or LB agar (Difco) was used for cloning and enumerating colony forming units (CFU) and supplemented with 10 μg/mL erythromycin, 10 μg/mL chloramphenicol, 100 μg/mL spectinomycin, 50 μg/mL gentamicin, 25 μg/mL irgasan, or 100-200 μg/mL carbenicillin as required. Salt-free LB + sucrose plates were made with 10 g/L Bacto-tryptone, 5 g/L yeast extract,10% v/v sucrose, and 15 g/L agar. Pyocyanin (from *Pseudomonas aeruginosa*, Sigma-Aldrich [P0046]) (PYO) was resuspended to a concentration of 10 mM in dimethyl sulfoxide (DMSO), stored at -30°C, and used at the concentration(s) indicated for each experiment.

### Pyocyanin and antimicrobial tolerance assays

Overnight cells were washed once in sterile PBS and normalized by their $OD_{600}$. Washed cells were then inoculated to an $OD_{600}$ of 0.1 in 1 mL of fresh, pre-warmed M63 (for PYO) or M63T (for the experiments in S4 Fig) and incubated for 2 hours in 14 mL polystyrene round-bottom test tubes. For the PYO tolerance assays shown in **Figs 1D**, **1E**, and S1 Fig, cells were inoculated at an $OD_{600}$ of 0.05 into 125 mL Erlenmeyer flasks containing 5 mL of fresh, pre-warmed M63 and grown to an $OD_{600}$ of ~0.25. After incubation, 1 mL of culture was transferred to 14 mL polystyrene round-bottom test tubes. PYO or the indicated antimicrobial was added at the indicated concentration (or the same volume of the respective solvent was added as a control: DMSO for PYO, HHQ (Sigma-Aldrich [SML0747-10MG]), 1-hydroxyphenazine (Tokyo Chemical Industry [H0289]), and phenazine-1-carboxylic acid (MolPort [001-738-598]),

ethanol for rhamnolipids (AGAE Technologies [R90-10G]), HQNO (Cayman Chemicals [15159]), and PQS (Sigma-Aldrich [94398-10MG]), or methanol for pyochelin (LGC Standards [TRC-P840365-1MG]), and pyoverdine (Sigma-Aldrich [P8124-1MG])) and the cells were incubated for an additional 20 hours. CFUs were enumerated by spot plating in triplicate. For experiments using hydrogen peroxide (Sigma-Aldrich [H1009]), paraquat (Fisher Scientific [US-PST-740]), carbonyl cyanide m-chlorophenyl hydrazone (CCCP; Sigma-Aldrich [215911-250MG]), or sodium arsenate (Sigma-Aldrich [S9663-50G]), the indicated supplement was added immediately prior to the addition of PYO.

## Laboratory evolution of PYO tolerance

Two independent populations of *S. aureus* JE2 were grown overnight in M63, washed in PBS, and inoculated to an $OD_{600}$ of 0.05 into 125 mL Erlenmeyer flasks containing 5 mL of fresh, pre-warmed M63 and grown to an $OD_{600}$ of ~0.25. 1 mL of culture was transferred to a sterile 14 mL test tube containing 20 µL of 10 mM PYO – yielding a final concentration of 200 µM PYO – and incubated for a further 20 hours. Following incubation, 20 µL of culture was removed to enumerate CFU and the remaining culture pelleted by centrifugation. Pelleted cells were resuspended in fresh M63 and allowed to recover overnight in the absence of PYO. Overnight recovered cells were then diluted as above, and the procedure repeated up to six times. Populations after treatment 5 for Population A and treatment 7 for Population B were streaked out on LB plates, and individual colonies were selected from these to test for PYO tolerance, and for whole-genome sequencing. We evolved two independent populations to identify common mutations that likely led to PYO adaptation. Given that we identified only one gene that was mutated in common in all sequenced isolates from both populations (*codY*), and *codY* mutations led to specific PYO tolerance, we did not perform additional PYO or media-only evolutions.

## Construction of *S. aureus* mutant strains

Primers used for cloning and verification are described in S3 Table. The locations of transposon insertions in strains acquired from the Nebraska Transposon Mutant Library (NTML) were verified by PCR and Sanger sequencing. Transductions were performed using φ11 or φ80 based on previously described procedures [120]. Briefly, overnight cultures bearing the transposon of interest were diluted 1:100 in a 125 mL Erlenmeyer flask containing 10 mL of BHI and grown for 2 hours ($OD_{600}$ ~1.0) at 37°C with shaking at 300 rpm. Then, cultures were supplemented with 150 µL of 1M $CaCl_2$ followed by the addition of 1-10 µL of empty phage, and incubation was continued overnight. Culture lysates were centrifuged for 10 minutes at ~4,000 rcf and the supernatant filtered with a 0.45 µm syringe filter. Subsequently, 100 to 1000 µL of phage lysate was used to transduce 1 mL of overnight culture of the recipient strain supplemented with 15 µL of 1M $CaCl_2$ in 15 mL conical tubes. Cultures were incubated as above for 20 minutes, at which point 200 µL of 200 µM sodium citrate was added, mixed by inversion, and centrifuged as above. Cells were then resuspended in BHI containing 1.7 mM sodium citrate, incubated as above for 1 hour, and then centrifuged as above. Transduced cells were concentrated 2-fold and 100-200 µL plated on at least three BHA plates containing 10 µg/mL erythromycin and 1.7 mM sodium citrate. Cells were allowed to grow at 37°C for up to 48 hours.

Cloning mutagenesis was performed using pIMAY* largely as previously described [121]. Briefly, the mutant *codY** allele was amplified with ~900+ bp of homology on each side and cloned into pIMAY*. ~1 ug of purified plasmid isolated from *E. coli* DC10B was transformed into electrocompetent *S. aureus* JE2 and directly selected for integration by incubation at 37°C as previously described [122]. Integration was confirmed following additional overnight

culture under antibiotic selection at 37°C using primers specific for plasmid integration. Integrated strains were then cultured at 28°C overnight for multiple passages without selection and plated on LB agar containing 20 mM para-chlorophenylalanine (PCPA) grown at 28°C. Chloramphenicol-sensitive colonies were screened by PCR and Sanger sequencing to identify the mutant allele.

## Construction of the *P. aeruginosa* Δ*phzM* mutant

Mutants were constructed as described previously [38]. Briefly, we amplified upstream and downstream fragments flanking the *phzM* gene, and an FRT-site flanked Gentamycin resistance cassette (from a pPS856 plasmid template [123]). All primer sequences are listed in S3 Table. We performed overlap extension PCR of these fragments to generate the deletion construct, which was cloned into the PCR8/GW/TOPO vector (Invitrogen) by TA cloning. The deletion construct was transferred from the PCR8/GW/TOPO-Δ*phzM* plasmid to the pEX18ApGW vector [123] using the LR Clonase II Enzyme mix (Invitrogen) for an LR reaction. The resulting pEX18ApGW-Δ*phzM* plasmid was transformed into the conjugative S17-1 λ-pir *E. coli* strain, and then introduced into the *P. aeruginosa* PA14 strain using bi-parental conjugation.

For conjugations, the donor and recipient strains were grown overnight in LB (the donor strain with the appropriate antibiotic), and 500 μL of each overnight culture was used for the conjugation. The cultures were washed twice with PBS and resuspended together in 100 μL of 100mM $MgSO_4$. This mixture was spotted on to LB plates, incubated at 37°C for 3 hours, scraped off with a plate spreader into PBS, and streaked on LB plates with the appropriate antibiotic to select for successful conjugants.

For the pEX18ApGW-Δ*phzM* plasmid, conjugants were selected on LB + irgasan + gentamicin plates, irgasan and gentamicin resistant colonies were selected on salt-free LB + 10% sucrose plates, and gentamicin-resistant carbenicillin-sensitive clones were tested for gene knockout by PCR and sequencing. A verified clone was conjugated with a S17-1 λ-pir *E. coli* strain carrying the pFLP2 plasmid that expresses the FLP recombinase [124], to excise out the gentamicin resistance cassette. Successful conjugants were selected on LB + irgasan + carbenicillin plates, irgasan and carbenicillin resistant colonies were selected on salt-free LB + 10% sucrose plates, and gentamicin and carbenicillin sensitive clones were tested for removal of the gentamicin cassette by PCR followed by sequencing.

## *S. aureus* – *P. aeruginosa* co-culture assays

Overnight grown *S. aureus* cells were washed once in sterile PBS, OD normalized and inoculated to an $OD_{600}$ of ~0.05 in 1 mL of fresh, pre-warmed M63 supplemented with 10 mg/mL tryptone, 4 mM $MgSO_4$, and 0.6% glycerol (M63T) in 14 mL polystyrene test tubes and grown for 4 hours at 37°C with shaking at 300 rpm. *P. aeruginosa* cultures were back-diluted in fresh M63T and grown as above for approximately 6 hours before dilution to an $OD_{600}$ of 0.1 in sterile PBS. *P. aeruginosa* was inoculated into 1 mL of *S. aureus* cultures at a final $OD_{600}$ of ~0.002 and grown as above for 18 hours. Viable *S. aureus* cells were enumerated on LB agar supplemented with 65 g/L sodium chloride to provide selection against *P. aeruginosa*.

## Pyocyanin growth curves

Overnight cells were washed once in sterile PBS, OD normalized and inoculated to an $OD_{600}$ of ~0.05 in fresh, pre-warmed M63. 100 μL of culture was aliquoted in duplicate to the wells of a 96-well plate. Plates were parafilmed to reduce evaporation and cells were grown for 20 hours with shaking (807 cpm) at 37°C in a BioTek Synergy H1 microplate reader (Agilent). Optical density values are adjusted by the background value at T0.

## Quantifying DNA damage using a TUNEL assay

Overnight cultures were washed and normalized as described above for the PYO tolerance assays. Washed cells were inoculated to a calculated $OD_{600}$ of 0.1 into 125 mL Erlenmeyer flasks containing 12 mL of fresh, pre-warmed M63 and incubated at 37°C for 2 hours with shaking at 300 rpm. After incubation, PYO at a final concentration of 200 µM or an equal volume of DMSO as a control was added and flasks were returned to the incubator. At the indicated time points, cells were pelleted by centrifugation at 14,400 rcf for 5 minutes, washed once with ice-cold PBS, and fixed on ice for 30 minutes in 1.5 mL of a 2.66% solution of PBS-paraformaldehyde (PFA). After initial fixation, cells were pelleted by centrifugation at 20,000 rcf for 3 minutes and washed with PBS to remove residual PFA. Washed cells were then resuspended in 1.25 mL of ice-cold 56% ethanol and stored for at least 24 hours at -30°C prior to TUNEL staining. TUNEL staining was performed using the APO-DIRECT Kits (BD Biosciences [51-6536AK, 51-6536BK]) according to the manufacturer's instructions. Stained cells were analyzed using an Apogee MicroPLUS flow cytometer (ApogeeFlow Systems Inc). *S. aureus* cells were gated using medium and large angle light scatter. Fluorescently labeled DNA was excited using a 488-nm laser and collected using 515 and 610 emission filters for FITC and propidium iodide, respectively, and analyzed using FlowJo (v10.1). Comparisons were made using the Overton method to identify the proportion of the population that exhibits fluorescence compared to the control condition.

## Whole-genome sequencing

Genomic DNA from *S. aureus* was isolated using a DNeasy Blood & Tissue Kit (Qiagen) with the addition of 5 µg/mL lysostaphin (Sigma) to the pretreatment regimen described for gram-positive bacteria. Sequencing libraries were prepared using the Illumina Nextera XT DNA Library Preparation Kit according to the manufacturer's instructions and sequenced by the CCR Genomics Core using a NextSeq 550 75 Cycle High Output kit for single-end sequencing, or the 150 Cycle Mid Output kit or High Output kit for paired-end sequencing. Genomes were assembled and mutations identified with breseq 0.33.1 [125] using the JE2 reference genome on NCBI (NZ_CP020619.1) as a reference.

## RNA-sequencing and analysis

Cells were cultivated in 125 mL Erlenmeyer flasks containing 10 mL of fresh, pre-warmed M63 medium at 37°C with shaking at 300 rpm. After the indicated treatment time bacterial RNA was stabilized using RNAprotect Bacteria Reagent (Qiagen) according to the manufacturer's instructions and stored at -80°C prior to RNA isolation. RNA was isolated using a Total RNA Plus Purification Kit (Norgen) with some modifications for *S. aureus*. Briefly, cryo-preserved bacterial pellets were resuspended in 100 µL of TE buffer containing 3 mg/mL lysozyme and 50 µg/mL lysostaphin and incubated for 30 minutes at 37°C. Volumes of Buffer RL and 95% ethanol used in the protocol were increased to 350 µL and 220 µL, respectively. Following elution of total RNA, any remaining genomic DNA was removed by TURBO DNase (Thermo Fisher) treatment using the two-step incubation method as detailed in the manufacturer's instructions. Removal of genomic DNA was confirmed by PCR.

Ribosomal RNA was then removed using Ribo-Zero rRNA Removal Kit for gram-positive Bacteria (Illumina) according to the manufacturer's instructions. Removal of rRNA was confirmed by electrophoresis using an Agilent TapeStation. RNA libraries were prepared using NEBNext Ultra II Directional RNA Library Prep Kit for Illumina (New England BioLabs) and sequenced using a NextSeq 550 75 Cycle High Output kit for single-end sequencing.

Sequence files were pre-processed using fastp [126]. Alignment was performed using Kallisto [127] and analyzed using EdgeR [128] and RStudio. Two independent RNA-seq experiments were performed and the replicate alignments were combined for analysis. Differential gene expression between conditions was performed with the glmQLFit function in EdgeR using an FDR significance of < 0.1 and a $\log_2$-fold cutoff of ≥ 1 or ≤ -1. Enriched pathways were identified using Gene Ontology Resource (www.geneontology.org) by the PANTHER Overrepresentation Test (released 20240226) (Annotation Version and Release Date: GO Ontology database https://doi.org/10.5281/zenodo.10536401Released2024-01-17) after conversion of differentially expressed JE2 locus tags to NCTC8325-4 using *Aureo*Wiki [129]. Pathway enrichment was tested using Fisher's exact test corrected for the false discovery rate. All scripts used to generate results and run the above programs are provided in S5 Data. Processed data files for each replicate are available in S6 Data.

## Statistics

Statistical analysis of data was performed using GraphPad Prism 9 (GraphPad Software, San Diego, CA, United States). Significance was determined by one-way or two-way analysis of variance (ANOVA), or a one-sample t-test as indicated in the figure legends. Log-scale values were log-transformed prior to statistical analysis.

## Supporting information

**S1 Fig.  PYO tolerance of experimental evolution isolates. PYO tolerance of terminal isolates from population A (A, B) and population B (C). Values indicate *S. aureus* viable cell counts after 20 hours of treatment with 200 μM PYO. The dashed lines indicate the mean initial cell density (CFU/mL) at the time of PYO addition. Data shown are the geometric mean ± geometric standard deviation of three biological replicates. Significance is indicated for comparison to the parent strain as determined by a one-way ANOVA using Dunnett's correction for multiple comparisons. (* *P* < 0.05, ** *P* < 0.01).**
(TIF)

**S2 Fig.  Growth of the *codY\** mutant in low concentrations of PYO is modestly greater than WT. Growth curves shown as $OD_{600}$ measurements of the WT and the *codY\** mutant in M63 containing (A) 0 μM PYO (DMSO control), (B) 12.5 μM PYO, (C) 25 μM PYO, and (D) 50 μM PYO. Data shown are the mean ± standard error of seven biological replicates.**
(TIF)

**S3 Fig.  A Δ*qsrR* mutant exhibits reduced tolerance to PYO. Viable cell counts are shown for the WT, and the *codY\** and Δ*qsrR* mutants after 20-hour treatment with the indicated concentration of PYO. Data shown are the geometric mean ± geometric standard deviation of four biological replicates. The upper, dashed line indicates the mean initial cell density (CFU/mL) for all strains and the lower, dotted line indicates the limit of detection. Significance is shown for comparisons to the respective WT condition, as determined by a two-way ANOVA using Dunnett's correction for multiple comparisons. (* *P* < 0.05, ** *P* < 0.01, *** *P* < 0.001, # *P* < 0.0001).**
(TIF)

**S4 Fig.  The *codY\** mutation confers tolerance to HQNO. Viable cell counts for WT and the *codY\** mutant following 20-hour treatment with (A) phenazines, (B) HHQ and PYO, (C) rhamnolipids and PQS, (D) HQNO, or (E) siderophores in M63T. Data shown are the geometric mean ± geometric standard deviation of three biological replicates. The dashed**

lines indicate the mean initial cell density (CFU/mL). Significance is shown for comparison between WT and the *codY\** mutant and was determined by a two-way ANOVA with row matching using Šídák's correction for multiple comparisons (* $P < 0.05$, ** $P < 0.01$, *** $P < 0.001$).
(TIF)

**S5 Fig. The transcriptional profile of the *codY\** mutant compared to WT is consistent with loss of CodY activity.** Differential gene expression of the *codY\** mutant compared to WT after 30 minutes of incubation in DMSO. (A) Enriched GO pathways from upregulated and downregulated genes are shown. (B) Volcano plot of $\log_2$(fold change gene expression) and -$\log_{10}$(false discovery rate). Upregulated genes are shown in light red and downregulated genes are shown in light blue. Individual genes from several pathways in (A) are further highlighted. A list of genes included in each pathway can be found in S3 Data.
(TIF)

**S6 Fig. Increased PYO tolerance conferred by the *codY\** mutation does not require agr activity.** Viable cell counts of WT, the *codY\** mutant, WT *agrA*::Tn, and a *codY\* agrA*::Tn double mutant after 20-hour treatment with 200 μM PYO. Data shown are the geometric mean ± geometric standard deviation of three biological replicates. The dashed lines indicate the mean initial cell density (CFU/mL) for the strains they specify. Significance is shown for a comparison to the respective WT background as determined by a one-way ANOVA using Šídák's correction for multiple comparisons. (*** $P < 0.001$, # $P < 0.0001$).
(TIF)

**S7 Fig. The response to PYO of the *codY\** mutant is similar to that of the WT but also exhibits metabolic suppression.** Transcriptional response of the *codY\** mutant to PYO compared to the DMSO control after (A, B) 30 and (C, D) 120 minutes. (A, C) Enriched GO pathways from upregulated and downregulated genes. (B, D) Volcano plot of $\log_2$(fold change gene expression) and -$\log_{10}$(false discovery rate). Highlighted genes comprise the pathways indicated in the figure legend. A list of genes in each pathway can be found in S3 Data.
(TIF)

**S8 Fig. Enriched pathways among the downregulated genes in the WT response to PYO after 120 minutes overlap significantly with that of the CodY regulon.** All enriched pathways among downregulated genes in the WT response to PYO compared to DMSO after 120 minutes.
(TIF)

**S9 Fig. Mutations in the *cidABCR* operon do not impact PYO survival in either the WT or *codY\** backgrounds.** Viable cell counts of WT, the *codY\** mutant, and their respective cid transposon mutants after 20-hour treatment with 200 μM PYO. Data shown are the geometric mean ± geometric standard deviation of three biological replicates. The dashed line indicates the mean initial cell density (CFU/mL) for all strains. Significance was determined relative to the isogenic parental (WT or *codY\**) strain by a two-way ANOVA using Dunnett's correction for multiple comparisons.
(TIF)

**S10 Fig. PYO induces DNA damage in *S. aureus*.** (A) Overton % positivity (reflecting the % of the population that has significant fluorescence compared to the control condition) after 2-, 4-, and 20-hour treatment with 200 μM PYO. (B) The difference in Overton

% positivity between the WT and *codY\** mutant in DMSO or PYO after the indicated treatment time. (A, B) Data shown are the mean ± standard deviation of three biological replicates. (C) Viable cell counts of WT after treatment with 200 μM PYO for the indicated time. Data shown are the geometric mean ± geometric standard deviation of four biological replicates. The dashed line indicates the mean initial cell density (CFU/mL). Significance is shown relative to (A) 0% Overton positivity, (B) the respective DMSO condition, or (C) the initial (0h) cell density as tested by a (A) one-sample t-test, (B) two-way ANOVA using Šídák's correction for multiple comparisons, or (C) one-way ANOVA using Dunnett's correction for multiple comparisons (# $P < 0.0001$). (D-F) Representative histograms from the data underlying (B) after (D) 2-, (E) 4-, and (F) 20-hour treatment.
(TIF)

**S11 Fig. Pathways enriched within genes differentially expressed in the *codY\** mutant compared to the WT in response to PYO are predominantly involved in amino acid metabolism.** Enriched GO pathways from upregulated and downregulated genes in the *codY\** response to PYO compared to the WT response to PYO after 30 (A) and 120 (B) minutes of treatment with PYO.
(TIF)

**S12 Fig. Excess CCCP sensitizes both WT and the *codY\** mutant to PYO.** Effect of CCCP addition (10 μM or 40μM) on survival after 20-hour treatment with either DMSO or 200 μM PYO for the (A) WT and (B) *codY\** mutant. Shown are the viable cell counts. (A, B) Data shown are the geometric mean ± geometric standard deviation of three biological replicates. Dashed lines indicate the mean initial cell density (CFU/mL) for all strains. Significance is shown for comparison to the indicated conditions and was determined by a one-way ANOVA using Tukey's correction for multiple comparisons. (\* $P < 0.05$, # $P < 0.0001$) .
(TIF)

**S13 Fig. Mutants deficient in the response to hydrogen peroxide stress are more sensitive to PYO while a mutant with a constitutive stress response phenocopies the *codY\** mutant.** Viable cell counts of WT, the *codY\** mutant, or the indicated transposon mutant strains after 20-hour treatment with 200 μM PYO. The dashed line indicates the mean initial cell density (CFU/mL) for all strains at the time of PYO addition. Data shown are the geometric mean ± geometric standard deviation of four biological replicates. Significance is indicated either for comparisons to the WT or between the indicated strains as determined by a one-way ANOVA using Dunnett's correction for multiple comparisons. (\* $P < 0.05$, \*\* $P < 0.01$, \*\*\* $P < 0.001$, # $P < 0.0001$).
(TIF)

**S14 Fig. WT and the *codY\** mutant are similarly susceptible to paraquat-derived superoxide.** Viable cell counts of WT and the *codY\** mutant after 20-hour treatment with the indicated concentration of paraquat. Data shown are the geometric mean ± geometric standard deviation of three biological replicates. The dashed line indicates the mean initial cell density (CFU/mL) for all strains. Significance was determined for comparison to the respective WT conditions by a two-way ANOVA using *Šídák's* correction for multiple comparisons.
(TIF)

**S15 Fig. Overexpression of *katA* confers tolerance to *P. aeruginosa* in co-culture.** Viable cell counts of *S. aureus* overexpressing *katA* or vector control co-cultured with *P. aeruginosa* PA14. Data shown are the geometric mean +/- geometric standard deviation

of three biological replicates. The dashed line indicates the mean initial cell density (CFU/mL) of both *S. aureus* strains prior to the addition of *P. aeruginosa*. Significance is shown for comparison between the two strains and was determined by a paired, two-tailed t-test. (** *P* < 0.01).
(TIF)

**S1 Table.  CodY-associated mutations in PYO-evolved isolates.**
(PDF)

**S2 Table.  Strains and plasmids used in this study.**
(PDF)

**S3 Table.  Oligonucleotides used in this study.**
(PDF)

**S1 Data.  A list of all mutations observed in each sequenced evolved isolate.**
(XLSX)

**S2 Data.  The full edgeR analysis, differentially expressed genes, and enriched PANTHER pathways for each RNA-seq comparison.**
(ZIP)

**S3 Data.  A list of genes comprising each pathway shown in the legend of the volcano plots.**
(XLSX)

**S4 Data  . CodY sequences of *S. aureus* isolates shown by position.**
(XLSX)

**S5 Data.  Scripts used for RNA-seq alignment and edgeR analysis.**
(ZIP)

**S6 Data.  Raw processed data files from kallisto in tsv format.**
(ZIP)

**S7 Data.  Numerical data underlying graphs and summary statistics.**

## (XLSX)

The authors declare no conflict of interest.

## Acknowledgments

We would like to acknowledge the Center for Cancer Research (CCR) Genomics Core for RNA-sequencing and whole-genome sequencing, and the Brinsmade lab (Georgetown University) for providing the pKM16 plasmid. We thank Susan Gottesman, Gisela Storz, Tiffany Zarrella, Kalinga Pavan Thushara Silva, and Stefan Katharios-Lanwermeyer for comments on the manuscript, and members of the Gottesman, Ramamurthi, and Khare labs for discussion and feedback throughout the study. This work used the computational resources of the NIH High Performance Computing Biowulf Cluster (http://hpc.nih.gov).

## Author contributions

**Conceptualization:** Anthony M. Martini, Sara A. Alexander, Anupama Khare.

**Data curation:** Anthony M. Martini, Sara A. Alexander, Anupama Khare.

**Formal analysis:** Anthony M. Martini, Sara A. Alexander, Anupama Khare.

**Funding acquisition:** Anupama Khare.

**Investigation:** Anthony M. Martini, Sara A. Alexander, Anupama Khare.

**Methodology:** Anthony M. Martini, Sara A. Alexander, Anupama Khare.

**Project administration:** Anupama Khare.

**Resources:** Anupama Khare.

**Supervision:** Anupama Khare.

**Visualization:** Anthony M. Martini, Sara A. Alexander, Anupama Khare.

**Writing – original draft:** Anthony M. Martini, Anupama Khare.

**Writing – review & editing:** Anthony M. Martini, Sara A. Alexander, Anupama Khare.

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
