## [Decision Letter · Decision Letter 0]

6 Feb 2025

Dear Dr Khare,

We are pleased to inform you that your manuscript entitled "Mutations in the Staphylococcus aureus Global Regulator CodY Confer Tolerance to an Interspecies Redox-Active Antimicrobial" has been editorially accepted for publication in PLOS Genetics. Congratulations!

Yours sincerely,

Jennifer Herman

Academic Editor

PLOS Genetics

Sean Crosson

Section Editor

PLOS Genetics

Aimée Dudley

Editor-in-Chief

PLOS Genetics

Anne Goriely

Editor-in-Chief

PLOS Genetics

Comments from the reviewers (if applicable):

Thank you very much for your patience during the review process and for choosing to resubmit your manuscript to PLOS Genetics. We are pleased to let you know the updated manuscript and responses to prior reviews have been uent evaluated by two experts in the field and only a very minor revision to text is requested. Congratulations and thank you again for your patience.

-Jen Herman

Reviewer's Responses to Questions

**Comments to the Authors:**

Reviewer #1: The authors have adequately addressed all my comments. The manuscript is greatly improved and I appreciate the authors’ efforts to streamline the narrative in the second part of their paper. I repeat my initial assessment: this is a great and thorough piece and I enjoyed reading it.

I have one small extra comment. The authors nicely explained the difference between resistance and tolerance (L122-124), but then later on (L161) they referred to “resistance” although they meant “tolerance”. Please check for consistency of terminology at the proof-correction stage.

Reviewer #2: This manuscript describes the effects of pyocyanin, a product made by Pseudomonas aeruginosa, on S. aureus experimental evolution. P. aeruginosa and S. aureus frequently co-infect. A pyocyanin tolerant CodY mutant also had a survival advantage in co-culture with P. aeruginosa, likely through tolerance specifically to pyocyanin. The increased tolerance of codY mutants contributed to increased production of catalase though metabolic adaptations also played a role and overproduction of KatA was sufficient to improve resistance to pyocyanin. Mutations predicted to cause loss of function in codY were also found in clinical isolates.

I appreciate the addition of the new experiments in response to previous comments and think that they strengthen the manuscript.

All data were appropriately deposited in public repositories.

**Have all data underlying the figures and results presented in the manuscript been provided?**

Reviewer #1: Yes

Reviewer #2: Yes

PLOS authors have the option to publish the peer review history of their article (what does this mean?). If published, this will include your full peer review and any attached files.

Reviewer #1: **Yes: **Rolf Kümmerli

Reviewer #2: No

**Data Deposition**

http://datadryad.org/submit?journalID=pgenetics&manu=PGENETICS-D-24-01433

**Press Queries**

---

## [Editor Report · Acceptance letter]

PGENETICS-D-24-01433

Mutations in the Staphylococcus aureus Global Regulator CodY Confer Tolerance to an Interspecies Redox-Active Antimicrobial

Dear Dr Khare,

We are pleased to inform you that your manuscript entitled "Mutations in the Staphylococcus aureus Global Regulator CodY Confer Tolerance to an Interspecies Redox-Active Antimicrobial" has been formally accepted for publication in PLOS Genetics! Your manuscript is now with our production department and you will be notified of the publication date in due course.

With kind regards,

Zsofia Freund

PLOS Genetics

On behalf of:
